# The Usefulness of X-ray Diffraction and Thermal Analysis to Study Dietary Supplements Containing Iron

**DOI:** 10.3390/molecules27010197

**Published:** 2021-12-29

**Authors:** Izabela Jendrzejewska, Robert Musioł, Tomasz Goryczka, Ewa Pietrasik, Joanna Klimontko, Josef Jampilek

**Affiliations:** 1Institute of Chemistry, University of Silesia, 40007 Katowice, Poland; ewa.pietrasik@us.edu.pl; 2Institute of Materials Science, University of Silesia, 40007 Katowice, Poland; tomasz.goryczka@us.edu.pl; 3Institute of Physics, University of Silesia, 40007 Katowice, Poland; joanna.klimontko@us.edu.pl; 4Department of Analytical Chemistry, Faculty of Natural Sciences, Comenius University, Ilkovicova 6, 842 15 Bratislava, Slovakia; josef.jampilek@gmail.com; 5Institute of Neuroimmunology, Slovak Academy of Sciences, Dubravska Cesta 9, 845 10 Bratislava, Slovakia

**Keywords:** dietary supplements, X-ray powder diffraction, qualitative phase analysis, thermal analysis

## Abstract

X-ray powder diffraction (XRPD) and thermal analysis (differential scanning calorimetry/derivative of thermogravimetry (DSC/DTG)) are solid-state techniques that can be successfully used to identify and quantify various chemical compounds in polycrystalline mixtures, such as dietary supplements or drugs. In this work, 31 dietary supplements available on the Polish market that contain iron compounds, namely iron gluconate, fumarate, bisglycinate, citrate and pyrophosphate, were evaluated. The aim of the work was to identify iron compounds declared by the manufacturer as food supplements and to try to verify compliance with the manufacturer’s claims. Studies performed by X-ray and thermal analysis confirmed that crystalline iron compounds (iron (II) gluconate, iron (II) fumarate), declared by the manufacturers, were present in the investigated dietary supplements. Iron (II) bisglycinate proved to be semi-crystalline. However, depending on the composition of the formulation, it was possible to identify this compound in the tested supplements. For amorphous iron compounds (iron (III) citrate and iron (III) pyrophosphate), the diffraction pattern does not have characteristic diffraction lines. Food supplements containing crystalline iron compounds have a melting point close to the melting point of pure iron compounds. The presence of excipients was found to affect the shapes and positions of the endothermic peaks significantly. Widening of endothermic peaks and changes in their position were observed, as well as exothermic peaks indicating crystallization of amorphous compounds. Weight loss was determined for all dietary supplements tested. Analysis of the DTG curves showed that the thermal decomposition of most food supplements takes place in several steps. The results obtained by a combination of both simple, relatively fast and reliable XRPD and DSC/DTG methods are helpful in determining phase composition, pharmaceutical abnormalities or by detecting the presence of the correct polymorphic form.

## 1. Introduction

Iron is one of the essential elemental nutrients involved in many key cellular processes. On the other hand, iron is responsible, for example, for oxidative damage to lipids, proteins or DNA [1]. The low-energy barrier between Fe(II)/Fe(III) oxidation states is responsible for crucial metabolic reactions, such as cellular respiration and oxygen transport, but also the formation of reactive oxygen species (ROS) [2]. Iron metabolism is therefore precisely regulated in the human body by various mechanisms. Due to impaired homeostasis, two conditions are possible: iron overload or deficiency. The first may result from frequent transfusions or genetic disorders [3]. Iron deficiency is a typical impact of malnutrition or parasitic infections. Iron deficiency can also occur in immunocompromised patients, anticancer therapy, various chronic diseases, and the elderly [3,4,5].

Although oral iron supplementation (OIS) is the first choice in cases of deficiency [6,7], it is essential to note that the bioavailability of iron is very problematic and depends on many factors [8,9]. Nevertheless, the OIS has risen dramatically recently; it is popular with patients for its low price and easy availability (number of over-the-counter products). The worldwide market for iron supplements has exceeded one million USD per year [10]. Given the processes involved in iron, it is important to assess needs and proper dosing to ensure the correct bioavailable iron concentrations. Dietary iron overload leads to non-hereditary hemochromatosis, in which high plasma iron levels affect hepatocytes, cardiomyocytes and pancreatic cells. Increased nutritional iron intake, such as high red meat consumption, is associated with a risk of diabetes or cancer [2,3,11]. In addition, recent reports suggest that increased iron levels and metabolic impairment of this element are factors in the severe symptoms of COVID-19 and complex therapies [12,13].

Supplements containing iron compounds or complexes are particularly demanding, as there are no simple, fast and reliable methods for determining the absolute value of Fe amount in the sample and, even more importantly, the oxidation state and co-ingredients of the sample matrix. In general, OIS preparations contain iron in two oxidation states Fe(II) and Fe(III), usually as salts or chelates. The preparations containing Fe(II) ions, so-called heme iron, are ideal for supplementation. The preparations with Fe(III) ions, called non-heme iron, are of low bioavailability and are ineffective. This opinion reflects well the situation in dietary iron uptake, where non-heme iron compounds come mainly from plant-based food packed with polyphenols, phytic acid salts and metals such as Ca or Mg, known as inhibitors of iron absorption [14,15]. However, OIS preparation that contains a much higher amount of iron and may be designed to fulfil specific requirements do not always comply with the condition mentioned above.

Moreover, the dietary supplements containing iron ions in the II oxidation state may have adverse effects and higher toxicity than more oxidized Fe(III). Toxicity significantly connected with ROS generation can be considerably elevated after chelation [16]. Compounds containing Fe(II) ions may change microbiota and are more often responsible for gastrointestinal tract issues like nausea, constipation, or diarrhea [17,18,19]. The nature of iron compounds, e.g., polymorphic form and particle size, and the nature of supplement, e.g., excipients and formulation, can affect bioavailability more than chemical composition or dose weight [20]. Formulations, e.g., addition of ascorbic acid, have a more substantial effect on absorption than the dose of FeSO_4_ or incubation time.

To avoid toxicity, Fe(III) (ferric) ions are often used in maltol or polymaltose complexes [21]. Fe(III) is particularly beneficial for patients with iron deficiency anemia who need long-term supplementation. Continuous OIS with ferrous (Fe(II)) compounds may disturb the copper-zinc homeostasis [22]. Specific formulations such as prolonged-release can be used to avoid at least some of these issues. The formulation is also crucial for stability issues as excipients originating from plant sources may affect the shelf life and final amount of iron form. It is known that ascorbic acid is capable of reducing both Fe(II) and Fe(III) [23]. Fruits or herbal extracts are very common adjuvants in supplements, regardless of their actual effectiveness.

Food supplements or special purpose foods form a “grey area” between the pharmaceutical market and other industries, characterized by a lack of systematic control, reliable reference materials and harmonized rules [24]. Rapidly developing market is driven by advertisements, product placement and blooming internet possibilities such as social media, influencers or pseudo-scientific information [25]. Another aspect that should be mentioned is the appearance of counterfeit supplements, expired or contaminated products and overestimated “snake oils”. It causes two risks for the consumer: (i) consumption of a harmful substance in interaction with other drugs or unsuitability for the patient, (ii) consumption of a sub-stance utterly devoid of any effect on the patient’s body.

Therefore, fast and efficient methods for evaluating OIS ingredients are needed to assess quality and safety; ideally, solid-state analytical techniques avoid any changes in composition during the analysis. This work continues our previous works focusing on the examination of dietary supplements and drugs using X-ray powder diffraction (XRPD) and differential scanning calorimetry (DSC) [26,27]. This work aimed to identify iron compounds declared by the manufacturer as food supplements and to determine whether the product meets the manufacturer’s claims. For the study, 31 popular and often-purchased dietary supplements containing iron compounds were selected from the Polish market. According to the main component of iron, they can be divided into iron (II) gluconate, iron (II) fumarate, iron (III) pyrophosphate, iron (II) bisglycinate and iron (III) citrate. Their formulations cover a single active ingredient and a complex preparation with adjuvants, plant or vegetable extracts (e.g., beetroot extract). Notably, the analysis of such complex formulations, which tend to be dark, not wholly soluble and prone to autocatalytic redox degradation, would be complicated using the approaches of classical liquid analytical chemistry.

## 2. Results and Discussion

### 2.1. X-ray Qualitative Phase Analysis

The X-ray qualitative phase analysis allows us to determine which crystalline phases are present in the investigated samples. Each polycrystalline phase gives a characteristic X-ray pattern with a specific position and intensity of diffraction lines. Moreover, each polycrystalline phase in the mixture provides its X-ray pattern, despite other phases coexisting. Therefore, such patterns may play the role of “fingerprints” in identifying substances. The identification technique consists in finding the phases whose reflections are consistent with the values for the tested material. The XRD method identifies the present phases at an amount below 1%. The X-ray detection limit is in the range of 0.1 to 1% wt. per phase, while the limit of detection (LOD) is approximately 1%. The details are described in [28,29]. These detection levels are suitable for typical compositions of OIS that can be found in the marker.

Qualitative X-ray analysis compares experimental diffraction data such as *2θ* diffraction angles, *d_hkl_* interplanar distances and relative intensities with the data from the ICDD database [30]. Values of *d_hkl_* interplanar distances were calculated based on the Bragg–Wulff equation [28]. Based on [31,32], it is assumed that the difference (|*∆2θ*|) in the position of the diffraction line (given a value of *2θ* angle) between the tested substance and the standard presented in the database should not be greater than 0.2°. A difference above 0.2° indicates that the tested product may be a counterfeit.

#### 2.1.1. X-ray Phase Analysis of Iron Compounds

Table 1 and Figure 1 present the results of XRPD analysis for iron (II) gluconate and iron (II) fumarate. The given crystal data have been obtained using the LeBail method.

Analysis of a diffraction pattern and obtained crystal data of pure iron (II) fumarate perfectly aligned with the PDF card 00-0062-1294 [30]. Similarly, crystal data and diffraction pattern of pure iron (II) gluconate are in good agreement with the data presented in [33]. The X-ray analysis was done for iron (II) gluconate monohydrate and iron (II) gluconate dihydrate. The diffraction pattern for both compounds is very similar. The XRPD method does not distinguish both forms of iron (II) gluconate. The other iron compounds included in the studied dietary supplements: iron (III) citrate and iron (III) pyrophosphate are amorphous substances. Structural parameters cannot be determined for these compounds. Although the diffraction pattern of iron (II) bisglycinate indicates amorphous properties, the diffractogram of the pure iron (II) bisglycinate showed a characteristic shape with two diffraction lines (Figure 2). The diffraction pattern of iron (II) bisglycinate indicated partly amorphous properties. The diffractogram of the pure iron (II) bisglycinate showed a characteristic shape with two diffraction lines (Figure 2). It allowed us to determine the values of *2θ* angles: 16.5449° and 22.6979° and to calculate the interplanar distance *d_hkl_*: 5.3536 Å and 3.9143 Å, respectively. Values of *d_hkl_* interplanar distances for all tested supplements were calculated based on the Bragg–Wulff equation [28].

#### 2.1.2. X-ray Analysis of Dietary Supplements Containing Iron (II) Gluconate

In Figure 3a,b, polycrystalline diffraction patterns of dietary supplements containing iron (II) gluconate are shown. Diffraction lines characteristic of iron (II) gluconate (C_12_H_24_FeO_14_) are present in all diffraction patterns.

Position of diffraction lines (*2θ* angles) and the calculated values of interplanar distances *d_hkl_* have values close to data obtained for pure iron (II) gluconate (Table 2). |*∆2θ*| values were lower than 0.2°, confirming the presence of iron (II) gluconate in studied dietary supplements. For some supplements, only deviations in intensities of diffraction lines were observed. The intensity of the registered diffraction line of the phase in the mixture depended on the content of a given phase, its crystalline structure, the nature of coexisting phases, and the crystallites’ orientation [27]. 

#### 2.1.3. X-ray Analysis of Dietary Supplements Containing Iron (II) Fumarate

Figure 4a,b show diffraction patterns of dietary supplements, which contain iron (II) fumarate as the main component delivering iron ions. Diffraction lines characteristic of iron (II) fumarate (C_4_H_2_FeO_4_) are depicted in all diffraction patterns. Most of the diffraction patterns also show diffraction lines derived from vitamin C. The positions of these lines are consistent with the PDF card 00-004-0308. The exception is the *Iron APTEO* supplement, which does not contain vitamin C, following the manufacturer’s declaration. |*∆2θ*| values were lower than 0.2°, confirming the presence of this component in the tested drugs (Table 3).

#### 2.1.4. X-ray Analysis of Dietary Supplements Containing Iron (II) Bisglycinate

Figure 5a–c show the diffraction patterns of dietary supplements containing iron (II) bisglycinate (C_4_H_8_FeN_2_O_4_) as the main ingredient. For most diffraction patterns, two diffraction lines characteristic for iron (II) bisglycinate are depicted. |*∆2θ*| takes values smaller than 0.2° confirming the presence of iron (II) bisglycinate in investigated supplements (Table 4). For the two supplements: *Chella Flex* and *Biofer*, the diffraction lines originating from iron (II) bisglycinate are hard to identify. The crystalline substances present in these two supplements appear as high-intensity diffraction lines. They are probably composed of grains with a privileged crystallographic direction. For this reason, the low-intensity diffraction lines derived from iron (II) bisglycinate are difficult to register and almost invisible in the diffraction pattern. Although these lines are feeble, we can notice their trace on diffraction pattern, which indicates the presence of iron (II) bisglycinate in the composition of investigated dietary supplements (Figure 5c). Diffraction lines derivating from vitamin C are in good agreement with the PDF card 00-004-0308.

#### 2.1.5. X-ray Analysis of Dietary Supplements Containing Iron (III) Pyrophosphate and Iron (III) Citrate

Because iron (III) pyrophosphate and iron (III) citrate are amorphous substances (Figure 2), it is impossible to identify these components using the XRPD method. The lack of clear diffraction lines characterizes the x-ray pattern of amorphous substances, and the counting intensity is very lower than for crystalline samples. There is a high background noise level concerning the maximum intensity. A particular procedure can be used to determine the amorphous amount in the mixture, for example, full-pattern fitting method with a simulated amorphous content (using Rietveld method).

It was only possible to identify the diffraction lines from the crystalline substances present in these supplements (Figure 6a–c). Diffractograms of tested dietary supplements showed diffraction lines derived from other components, for example, sorbitol, vitamin C, cellulose or TiO_2_. The lines visible for the diffraction pattern of dietary supplements with iron (III) pyrophosphate, derived from sorbitol, were lined with the PDF 00-039-1923 card, while those with the vitamin C were lined with the PDF 00-004-0308 card. On the diffraction pattern for dietary supplements: *Iron citrate Swanson*, the widened line between 20–25° of *2θ* indicates cellulose. The strongest diffraction line (*2θ* = 25.61°) confirmed the TiO_2_ presence (Figure 6c). A similar phenomenon, confirming cellulose’s presence, is observed for other dietary supplements, for example, *Iron APTEO*, *Ferr C*, *Restonum LS* or *Organic Iron Aliness*.

On diffractograms, unidentified lines are observed. It is probably because the crystalline phase detectability depends on the mixture type in which this phase occurs. It is related to the change of the absorption coefficient ratio of the phase and the mixture as a whole $\frac{\mu_{j}^{*}}{\mu^{*}}\;(\mu_{j}^{*}$—mass absorption coefficient of the phase *j*, $\mu^{*}$—mass coefficient of the mixture) and the overlapping of the reflections of the coexisting phases.

### 2.2. Thermal Analysis

Thermal measurements using DSC/TG (DSC—Differential Scanning Calorimetry, TG—Thermogravimetry) were carried out for pure iron compounds and 21 selected dietary supplements. The technical details of these measurements are described in [27]. The remaining samples of dietary supplements exploded during the measurement and posed a threat to the equipment. The DSC/TG data are presented in Table 5, Table 6, Table 7, Table 8 and Table 9, while only the temperatures connected with DTG maximum are shown in figures. On the TG curve, steps related to the change in mass during heating or cooling could be observed. These steps are often blurred and may overlap when several reactions occur in a sample in succession. A DTG curve can be generated based on the TG curve, making it easier to distinguish overlapping thermal effects and accurately determine each reaction’s start and end temperature associated with a mass change.

#### 2.2.1. Thermal Analysis of Iron Compounds

Figure 7a,b show results of thermal analysis of pure iron compounds present in dietary supplements. Table 5 presents thermal parameters for the examined iron compounds obtained by DSC/TG.

The DSC curves of iron compounds showed endothermic peaks indicating the tested samples’ ongoing melting processes and thermal decomposition. One endothermic peak was observed for iron (II) fumarate at T = 460.6 °C, indicating thermal decomposition. The TG/DTG curve for pure iron (II) fumarate shows its stability up to a temperature of 460.6 °C. At this temperature, the carbonic matter decomposes into gaseous products indicating that the thermal composition was complete (Figure 7a).

Although the X-ray analysis does not distinguish the hydrated form of iron (II) gluconate, the DSC/TG analysis of pure iron (II) gluconate monohydrate and iron (II) gluconate dehydrate has shown apparent differences (Figure 7b). For iron (II) gluconate monohydrate, the first endothermal peak caused by dehydration occurred at 138.8 °C. The second endothermal peak occurred at 189.9 °C, pointing to the thermal decomposition of iron (II) gluconate monohydrate (Table 5). For iron (II) gluconate dihydrate, the first endothermal peak connecting with dehydration occurred at 148.3 °C. The second endothermal peak occurred at 195.3 °C, indicating the thermal decomposition of iron (II) gluconate dihydrate (Table 5). DTG curves suggest that, for both compounds, the thermal decomposition takes place at the same temperature ~200 °C (Figure 7b). The loss of weight is more significant for iron (II) gluconate monohydrate (Table 5). 

DSC data for iron (III) pyrophosphate are not available in the literature. Still, our investigations have shown an endothermic peak at 275.0 °C, which is in good accordance with the DTG minimum (Table 5, Figure 7a). The rate of decomposition is the fastest at this temperature.

Similarly, the melting point of iron (III) citrate is not found in the literature. Still, our measurements showed the presence of two endothermic peaks, the first at 201.9 °C, and the second at 332.2 °C. DTG curve indicate that the decomposition of this sample took place in two stages, the fastest at 203.2 °C. For iron (II) bisglycinate, the endothermic peaks were not observed on the DSC curve (Figure 7b). However, the TG/DTG curves indicate the process of decomposition, which takes place in several steps. 

#### 2.2.2. Thermal Analysis of Dietary Supplements Containing Iron (II) Gluconate

Figure 8 shows the results of thermal analysis of dietary supplements containing iron (II) gluconate. In this group of dietary supplements, it was observed that, for the *Ascofer*, *Fe-Miron* and *Ferrous gluconate Puritan’s Pride* supplements, the shape of the DSC curve is close to the DSC curve for pure iron (II) gluconate monohydrate. It was observed that the first endothermic peak appeared in the temperature range 137–140 °C, while the second endothermic peak from iron (II) gluconate was visible in the temperature range 181–194 °C. For the *Floridax* supplement, no endothermic peaks were observed, only a weight loss (TG curve) of approx. 70% was observed. The DTG curve of *Floridax* is in line with the DTG curve of *Ascofer* (Figure 8). Both supplements decompose at ~200 °C. 

On the DSC curve for *Fe-Miron*, the exothermic peak was observed at ~340 °C. It suggests the crystallisation process of amorphous substances present in their composition. The presence of an amorphous substance is confirmed by an increased background in the diffraction pattern (Figure 3a). Further increase of temperature causes mass loss and complete decomposition of investigated supplements. 

The endothermic peaks at 58.42 °C and 59.11 °C observed on the DSC curves for *Ascofer* and *Fe-Miron*, respectively, confirm the presence of fatty acids declared by the producers (Table 6). The DTG curves of investigated supplements suggest that the decomposition process runs in several steps.

#### 2.2.3. Thermal Analysis of Dietary Supplements Containing Iron (II) Fumarate

Figure 9 and Table 7 present the results of DSC/TG measurements for dietary supplements containing iron (II) fumarate: *Ferr C Bicaps*, *Restonum LS*, *Iron APTEO* and *Organic Iron Aliness*.

The small endothermic peaks were observed on the DSC curves at a temperature close to one for the pure iron (II) fumarate, i.e., 450 °C. The position of these peaks is in line with the data for pure iron (II) fumarate (Table 5). However, the DSC shapes for investigated dietary supplements contain endothermic peaks from other substances included in the tested supplements. In the mixture, the maximum endothermic peaks are obtained for different temperatures than for pure substances. The change in the melting point of additional substances in a combination depends on its amount and, more precisely, on its mole fraction value.

It should be noted that the shape of the TG/DTG curves indicate that the thermal decomposition of the tested supplements occurs in several stages. For all samples, the last minimum in the DTG curve appears at about 450 °C. It points to the complete decomposition of these supplements at a temperature close to the pure iron fumarate—450 °C (Table 5). Exothermic peaks visible in the DSC curves may indicate crystallisation of amorphous components in the tested supplements. It is also evidenced by the increased background in their diffraction image (Figure 4a).

#### 2.2.4. Thermal Analysis of Dietary Supplements Containing Iron (III) Pyrophosphate

Although DSC analysis revealed a small endothermic peak at 275 °C, for iron (III) pyrophosphate, on the DSC curves for dietary supplements, endothermic peaks originate from other substances were observed (Figure 10); it can be seen in Table 8 that, for *Actiferol*, *Dicofer Junior* and *SiderAl Forte*, the maximum endothermic peak occurs in the range of 84–95 °C, confirming the presence of sorbitol (m.p. ≈ 95 °C), which, according to the manufacturer’s declaration, is in good accordance with diffraction pattern (Figure 6a–c).

The TG/DTG curves indicate that the decomposition temperature for investigated supplements took place in several steps. The exception is *SiderAl Forte*, for which the process of thermal decomposition took place only in one step, by about 290 °C. A slightly different phase composition may cause the different behavior of *SiderAl Forte* compared to other supplements, especially *SiderAl Folic*. The shape of the DTG curves for the two supplements was similar, although a split into two peaks is visible for *SiderAl Folic*. The presence of other substances (e.g., B vitamins, vitamin D) in this supplement causes decomposition in two stages. The *SiderAl Forte* supplement does not contain these vitamins in its composition, hence the different shape of the DTG curve, indicating that the decomposition of this supplement takes place immediately at the temperature of ~290 °C.

#### 2.2.5. Thermal Analysis of Dietary Supplements Containing Iron (II) Bisglycinate and Iron (III) Citrate

The dietary supplements containing iron (II) bisglycinate: Balanced *Iron Viridian*, *Biofer*, *Chella Ferr Forte*, *Chella Ferr Med*, *Chella Flex*, *Ferrum DOZ*, *Szelazo SR*, *X-Ferr*, *Gentle Iron* and *Iron NOW* have been measured using DSC/TG method. The results are presented in Figure 11a–c, and in Table 9.

The shape of the DSC curves is different from the DSC curve for pure iron (II) bisglycinate. It is connected with the amorphous state of iron (II) bisglycinate and other substances on the composition of dietary supplements. For dietary supplements, *Chella Flex*, *X-Ferr*, *Chella Ferr Forte*, *Chella Ferr Med*, *Gentle Iron*, *Iron NOW*, endothermic peaks are visible on DSC curves, for which thermal parameters could be determined (Table 9). These peaks originate from crystalline substances contained in dietary supplements. For all tested dietary supplements containing iron (II) bisglycinate, the TG curves showed a weight loss in the range of 50–60%. The beginning of the sample melting process can be found in the DTG curves. In most cases, the melting process began at lower temperatures. It occurred in several steps, except for supplements *Szelazo SR*, *Iron DOZ* and *Balanced Iron Viridian*, where only one peak is visible in the DTG curves, indicating the melting point of the investigated supplement.

The DSC curve for dietary supplements *Iron citrate Swanson* shows two endothermic peaks (Figure 12). These peaks occur at different temperatures in comparison with the pure iron (III) citrate. It was probably connected with the presence of other crystalline substances in the composition of this supplement. It is in good agreement with the diffraction pattern for *Iron citrate Swanson* (Figure 6c).

## 3. Materials and Methods

### 3.1. Materials

The research was conducted for 31 dietary supplements containing iron ions in various compounds: iron (II) gluconate, iron (II) fumarate, iron (II) bisglycinate, iron (III) pyrophosphate and iron (III) citrate. They can be purchased at a pharmacy, store or online. All tested drugs are summarized in Table 10. The table includes the data declared by the manufacturer: the type and amount of iron compounds. Pure iron compounds were used for comparative studies: iron (II) gluconate monohydrate (Merck Life Science, Darmstadt, Germany), iron (II) gluconate dihydrate (Merck), iron (II) fumarate (Merck), iron (III) pyrophosphate (Merck) and iron (III) citrate (Merck). Iron (II) bisglycinate was isolated from the X-Ferr (Galena) supplement.

Excipients such as starch, cellulose, ascorbic acid, etc., are omitted in the table because the primary purpose was to identify iron compounds as the main substance in the investigated dietary supplements. However, for some supplements, the diffraction lines from declared other substances were determined due to the high intensity of these lines.

### 3.2. Methods

All dietary supplements and standard substances were studied using X-ray analysis (XRPD—X-ray Powder Diffraction) and thermal analysis (DSC/TGA, DSC—Differential Scanning Calorimetry, TGA—Thermogravimetry). X-ray analysis was performed using D5000 (Siemens, Munich, Germany) and PW3050 (X’Pert Philips, Malvern Panalytical, Malvern, UK) polycrystalline diffractometers. All samples were ground thoroughly in an agate mortar to obtain a high degree of homogeneity. The values of the interplanar distances *d_hkl_* were calculated based on the Bragg Equation (1):(1)$$d_{hkl}=\frac{n\lambda }{2\sin\theta }$$
where: *d_hkl_*—interplanar distance, *n*—reflection order, *λ*—wavelength, *θ*—angle of reflection.

Because the strongest diffraction lines are observed at low angles in the obtained diffraction patterns, all diffraction images are presented for the angular range *2θ*: 5–40°. The diffractometric data from the ICDD PDF4 database (Table 11) was used to compare the experimental data with the standards. The DSC/TG measurements were carried out using Setaram Thermo analyser *Labsys Evo*. The XRD and DSC/TG methods and their technical details are described in [27].

## 4. Conclusions

The XRPD made it possible to state that the active ingredients declared by the manufacturers in their specifications were present in the analyzed products containing crystalline and semi-crystalline iron compounds iron (III) pyrophosphate and iron (III) citrate, as amorphous substances do not give a clear diffraction pattern. XRPD cannot verify the presence of these compounds in the tested supplements. Analysis of the DSC and DTG curves showed that the OIS containing crystalline iron compounds had a melting point close to the melting point of the pure compounds. DSC/DTG studies for iron (II) gluconate monohydrate and iron (II) gluconate dihydrate distinguished these hydrates and confirmed that iron (II) gluconate monohydrate was present in the tested dietary supplements. Based on the DTG curves, it can be concluded that all tested OIS containing iron (II) gluconate are subject to thermal decomposition in the temperature range 180–200 °C.

Similarly, preparations containing iron (II) fumarate mixed with other crystalline bulk excipients in other compounds. Therefore, several endothermic peaks were observed in the DSC curves. These supplements thermally decomposed at a temperature of about 450 °C, which is visible on the DTG curves and is in good agreement with the decomposition temperature of pure iron fumarate (~460 °C). DSC curves for amorphous iron (III) citrate and iron (III) pyrophosphate showed small endothermic peaks, consistent with DTG curves, indicating an ongoing decomposition reaction. Dietary supplements containing amorphous iron compounds also contained crystalline substances. The endothermic peaks observed in the DSC curves were derived from these components. DTG curves show that decomposition occurs in several stages depending on the supplement’s composition. These observations were consistent with XRPD analysis, revealing many diffraction lines derived from crystalline substances. The DSC curves of the tested supplements showed exothermic peaks, indicating the presence of an amorphous phase that crystallizes under high temperatures. The weight loss observed in the TG curves and the changes in the DTG curves confirm the ongoing decomposition reactions. The DSC curve for iron (II) bisglycinate showed the presence of an amorphous component (visible broad exothermic peak). Vast endothermic peaks were also visible on the DSC curves of the tested OIS, confirming the existence of this compound in the supplement composition. Table 12 clearly provides basic observations. The specific findings are as follows: for supplements **1**–**6** (crystalline iron (II) gluconate) and supplements **7**–**13** (crystalline iron (II) fumarate) the iron compounds were proved by XPRD, as well as by thermal analysis, while in supplements **19**–**30** (amorphous iron (II) bisglycinate) and supplement **31** (amorphous iron (III) citrate), the iron compounds were not possible to determine by XRPD but were proved by thermal analysis. Iron compounds in supplements **14**–**18** (semicrystalline iron (III) pyrophosphate) were not clearly detectable by XPRD but were proved by thermal analysis. Based on research performed by thermoanalytical and X-ray techniques, it can be concluded that the combination of both methods can determine the phase composition of iron supplements and can especially confirm the presence of the main component. The choice of the behavior of the tested OIS at different temperatures allowed for defining the temperature limits below which the analyzed substances can be processed without changing their physicochemical properties. The obtained results can also be helpful as a guide for testing the stability of dietary supplements and can be used to detect inconsistencies in their phase composition. It can be stated that the results obtained from XRPD and DSC/DTG measurements correlate well with each other.

## Figures and Tables

**Figure 1 molecules-27-00197-f001:**
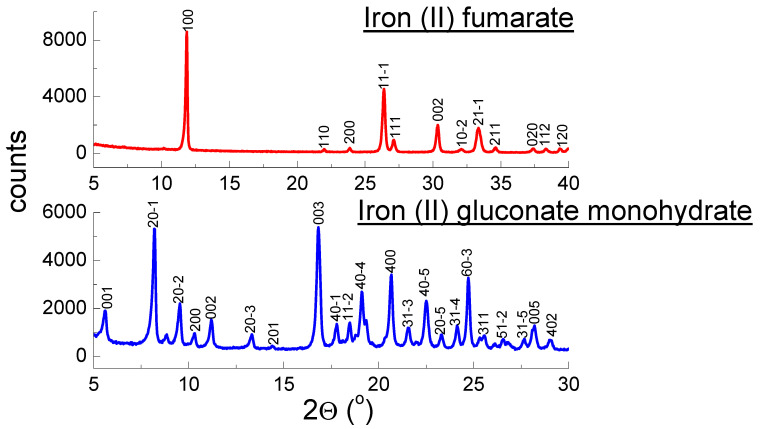
Diffraction patterns of iron (II) gluconate monohydrate and iron (II) fumarate.

**Figure 2 molecules-27-00197-f002:**
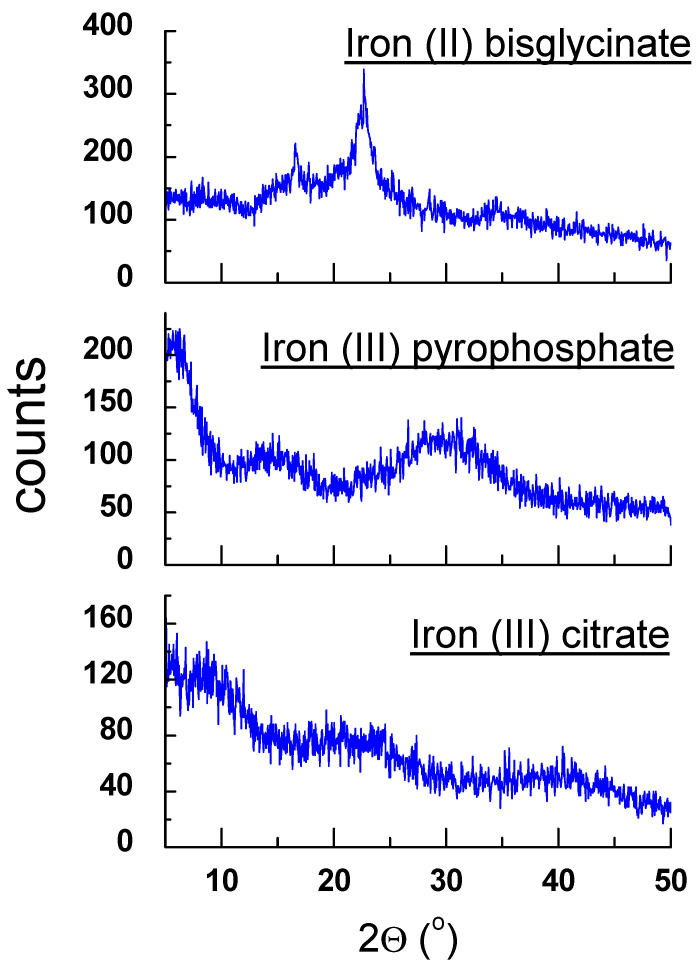
Diffraction patterns of iron (II) bisglycinate, iron (III) pyrophosphate and iron (III) citrate.

**Figure 3 molecules-27-00197-f003:**
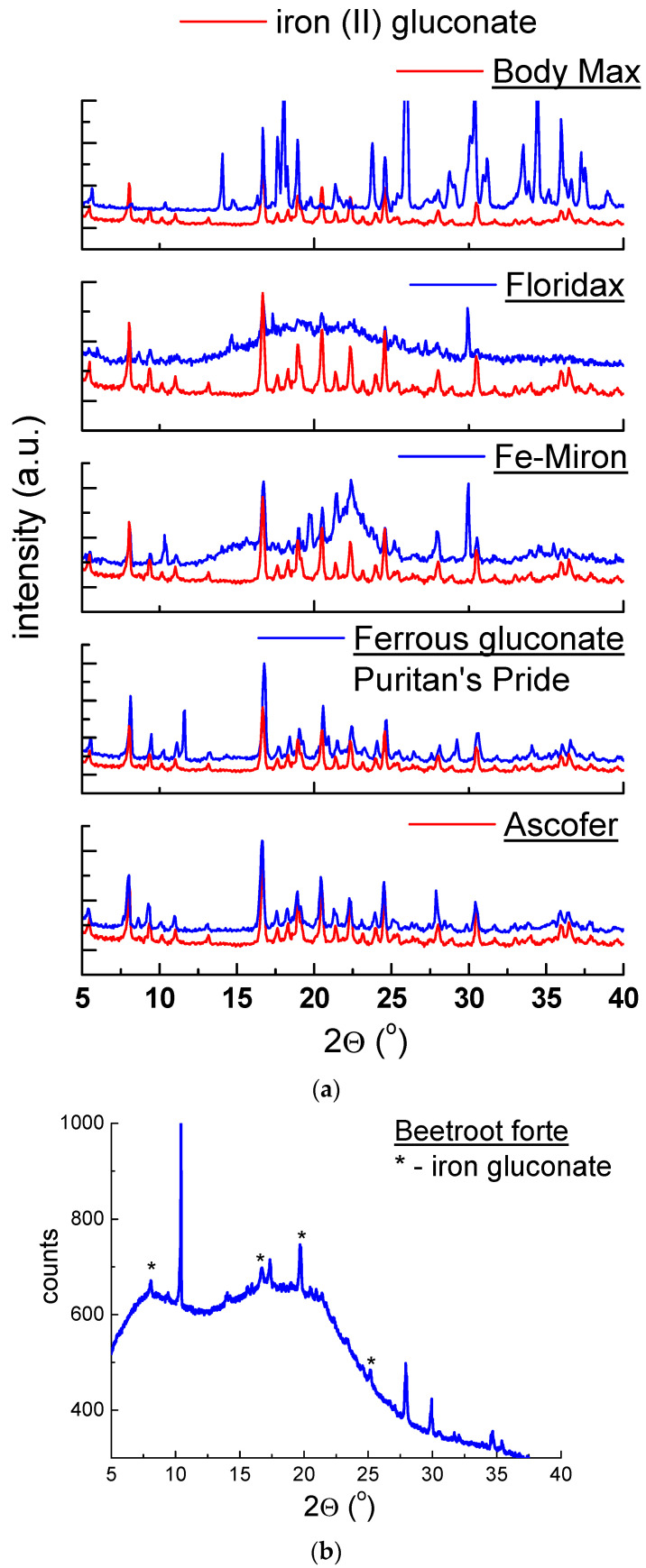
Diffraction patterns for dietary supplements: (**a**) *Ascofer*, *Ferrous gluconate Puritan’s Pride*, *Fe-Miron*, *Floridax*, *Body Max* and (**b**) *Beetroot forte*.

**Figure 4 molecules-27-00197-f004:**
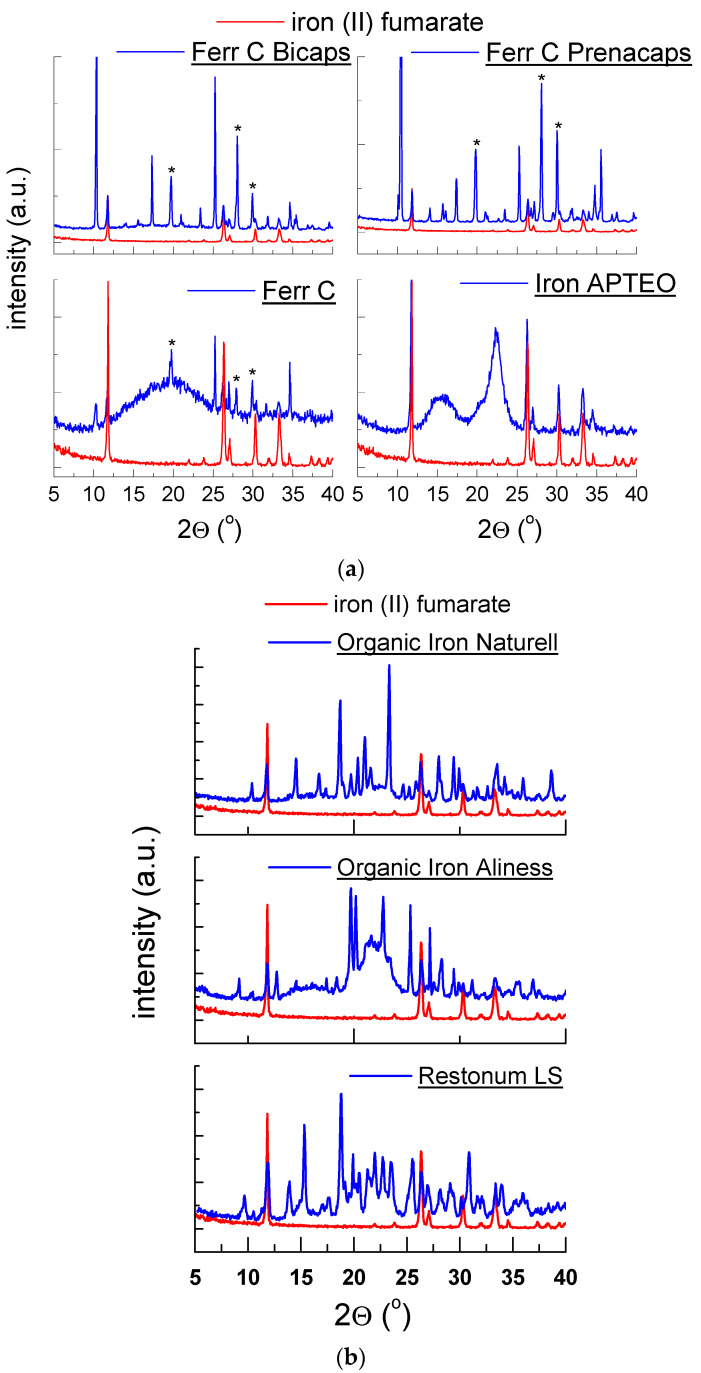
Diffraction patterns for dietary supplements: (**a**) *Ferr C Bicaps*, *Ferr C Prenacaps*, *Ferr C*, *Iron APTEO*, (**b**) *Restonum LS*, *Organic Iron Naturell*, *Organic Iron Aliness*, *—vitamin C.

**Figure 5 molecules-27-00197-f005:**
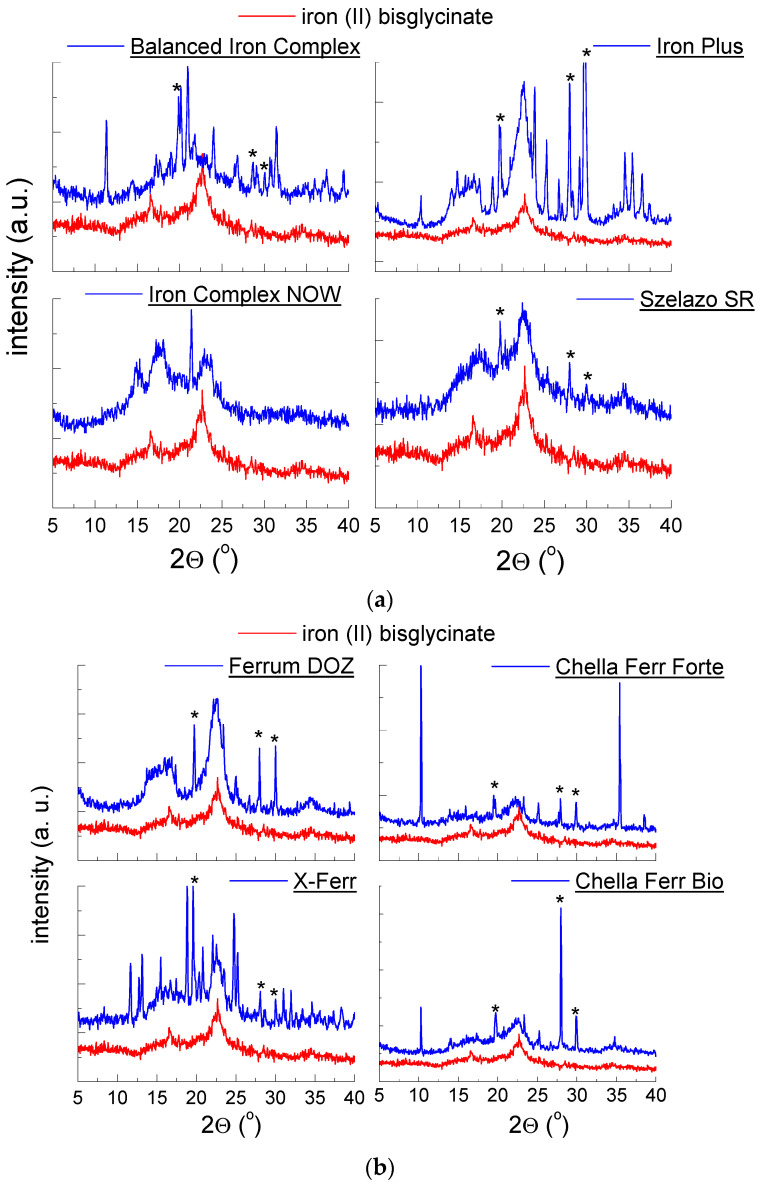
Diffraction patterns for dietary supplements: (**a**) *Balanced Iron Complex*, *Iron Plus*, *Iron Complex Now*, *Szelazo SR*, (**b**) *Ferrum DOZ*, *Chella Ferr Forte*, *X-Ferr*, *Chella Ferr Bio*, (**c**) *Biofer*, *Chella Ferr Med*, *Chella Flex*, *Gentle Iron*, *—vitamin C.

**Figure 6 molecules-27-00197-f006:**
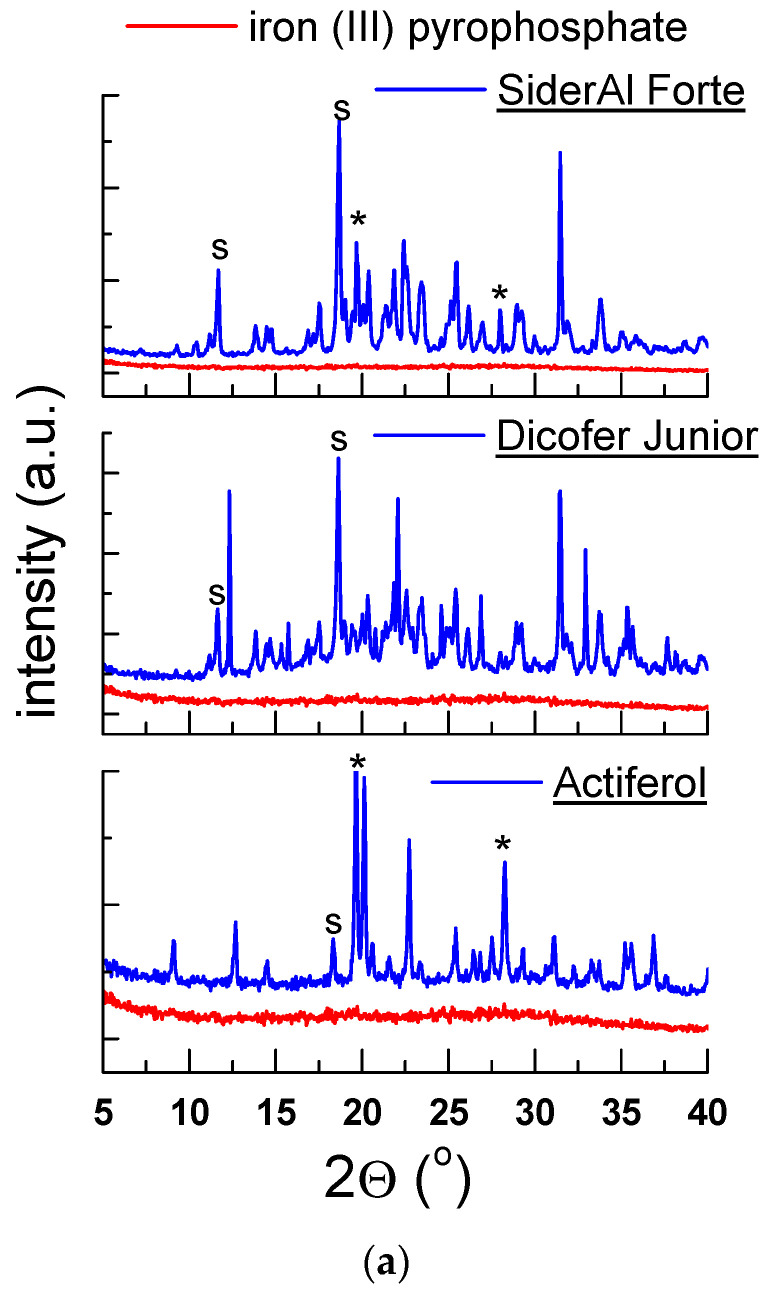
Diffraction pattern for dietary supplements: (**a**) *SiderAl Folic*, *Dicofer Junior*, *Actiferol*, (**b**) *SiderAl Folic*, *SmartHit IV Ferrum*, (**c**) *Iron citrate Swanson*, s—sorbitol, t—TiO_2_, *—vitamin C.

**Figure 7 molecules-27-00197-f007:**
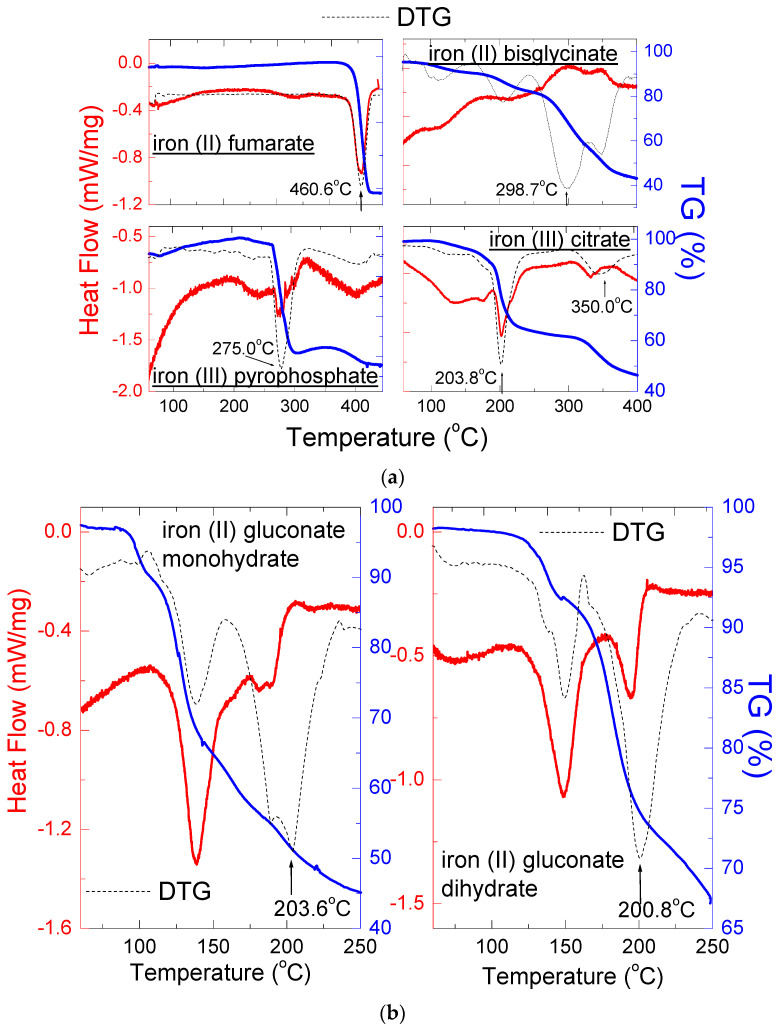
DSC and TG/DTG curves of pure iron compounds: iron (II) fumarate, iron (III) pyrophosphate, iron (II) bisglycinate and iron (III) citrate (**a**) and for iron (II) gluconate monohydrate and iron (II) gluconate dihydrate (**b**).

**Figure 8 molecules-27-00197-f008:**
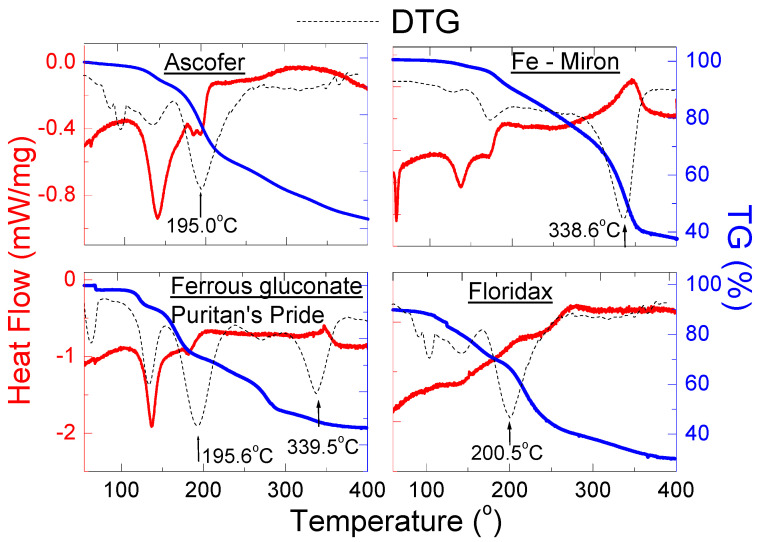
DSC and TG/DTG curves of dietary supplements containing iron (II) gluconate: *Ascofer*, *Fe-Miron*, *Iron gluconate Puritan’s Pride* and *Floridax*.

**Figure 9 molecules-27-00197-f009:**
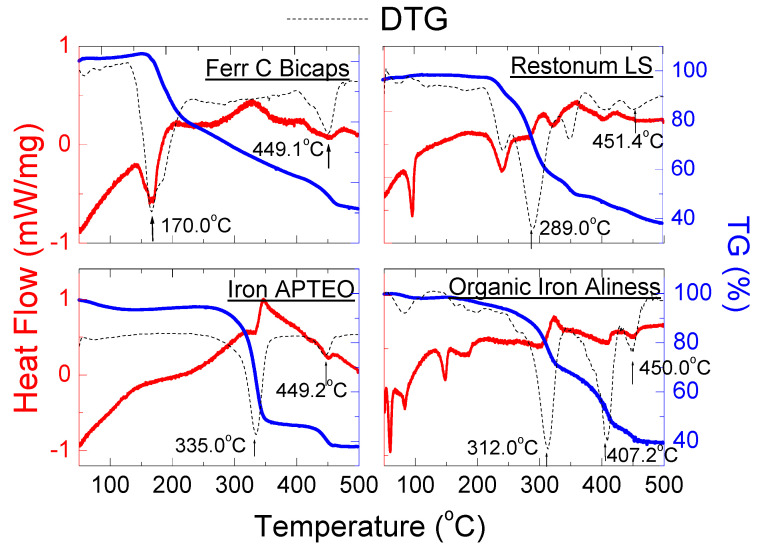
DSC and TG/DTG curves of dietary supplements containing iron (II) fumarate: *Ferr C Bicaps*, *Restonum LS*, *Iron APTEO* and *Organic Iron Aliness*.

**Figure 10 molecules-27-00197-f010:**
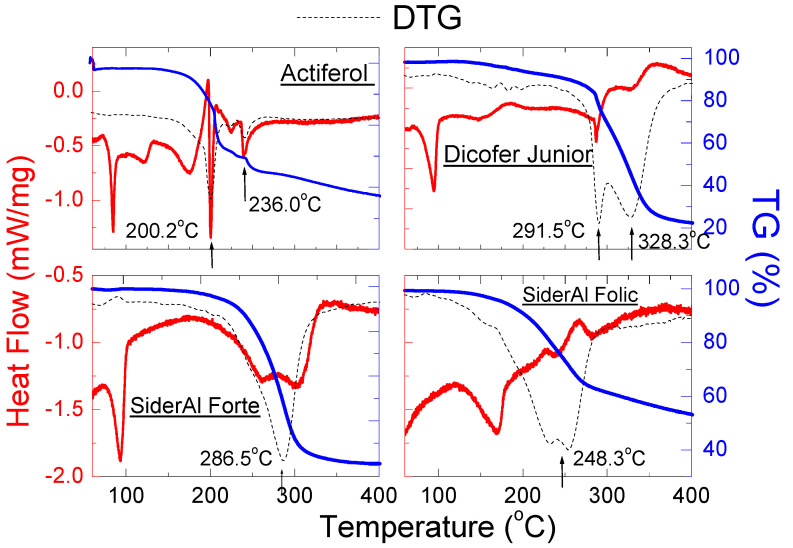
DSC and TG/DTG curves of dietary supplements containing iron (III) pyrophosphate: *Actiferol*, *Dicofer Junior*, *SiderAl Folic* and *SiderAl Forte*.

**Figure 11 molecules-27-00197-f011:**
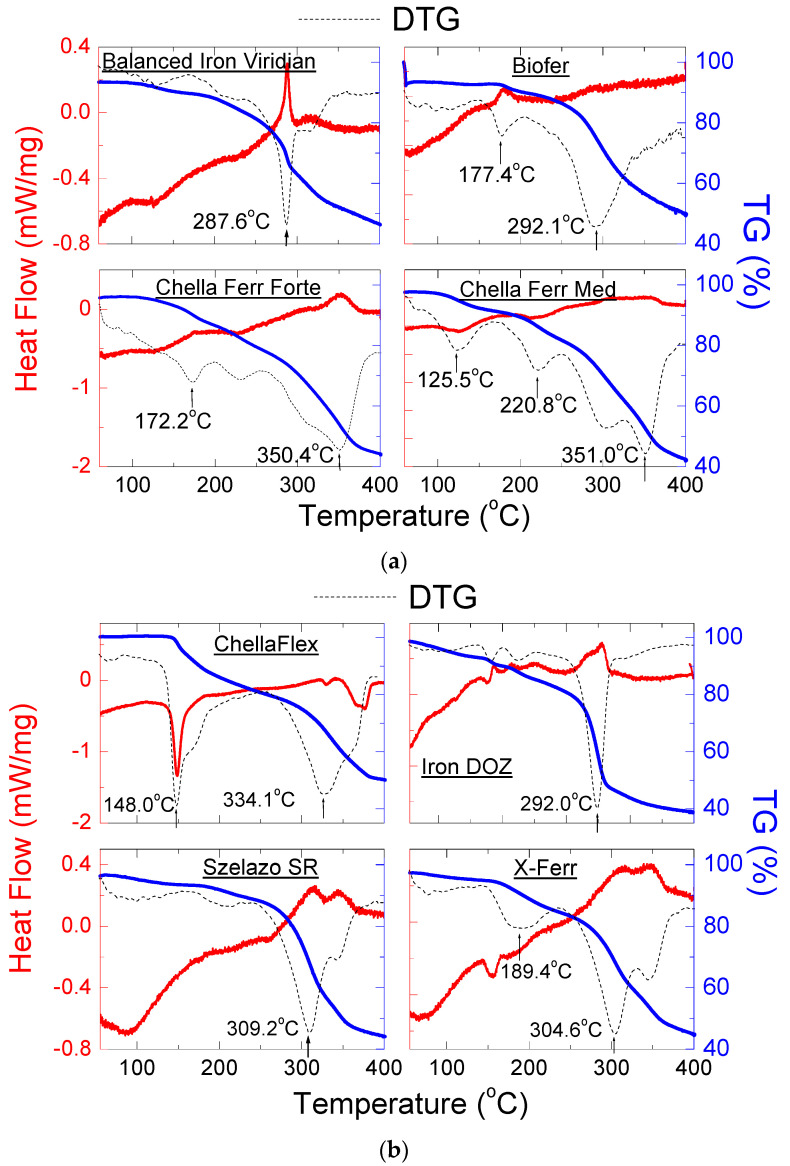
DSC and TG/DTG curves of dietary supplements containing iron (II) bisglycinate: (**a**) *Balanced Iron Viridian*, *Biofer*, *Chella Ferr Forte*, *Chella Ferr Med*, (**b**) *Chella Flex*, *Iron APTEO*, *Szelazo SR*, *X-Ferr*, (**c**) *Gentle Iron*, *Iron NOW*.

**Figure 12 molecules-27-00197-f012:**
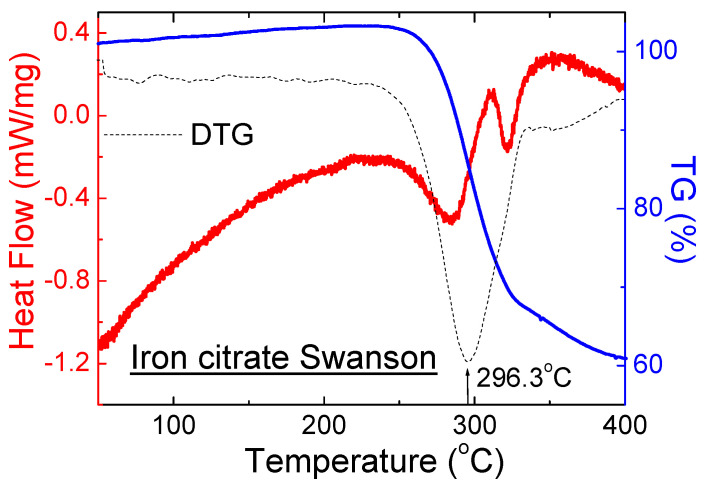
DSC and TG/DTG curves of a dietary supplement containing iron citrate: *Iron citrate Swanson*.

**Table 1 molecules-27-00197-t001:** The crystal data obtained for iron (II) gluconate and iron (II) fumarate.

Crystal Data	Iron (II) Gluconate		Iron (II) Fumarate	
Formula, Z	C_12_H_24_FeO_14_, Z = 4		C_4_H_2_FeO_4_, Z = 2	
Molecular Mass (g/mol)	448.16		169.90	
Symmetry, Space group	Monoclinic, I2 (5)		Monoclinic, P2_1_/c (14)	
Structural parameters	a = 19.9582(2)Å	α = 90°	a = 7.4646(3)Å	α = 90°
	b = 5.5089(3)Å	β = 111.37(1)°	b = 4.8022(3)Å	β = 92.99(1)°
	c = 18.4773(2)Å	γ = 90°	c = 5.8976(2)Å	γ = 90°
Volume (Å^3^)	1891.35		211.11	
Density (g/cm^3^)	1.574		2.673	

**Table 2 molecules-27-00197-t002:** Comparison of experimental data of dietary supplements containing iron (II) gluconate with the experimental data for pure iron (II) gluconate.

No. of Diffraction Line	*2θ* (°) ds *	*2θ* (°) ig **	|*∆2θ*|	*d_hkl_* (Å) ds *	*d_hkl_* (Å) ig **
*Ascofer*					
1.	8.1906	8.1845	0.0061	10.7859	10.7940
2.	16.8808	16.8312	0.0496	5.2478	5.2632
3.	19.1196	19.1253	0.0057	4.6381	4.6367
4.	20.6338	20.6515	0.0177	4.3010	4.2974
5.	24.7274	24.7242	0.0032	3.5975	3.5979
*Ferrous gluconate Puritan’s Pride*					
1.	8.1550	8.1845	0.0295	10.8329	10.7940
2.	16.7948	16.8312	0.0264	5.2745	5.2632
3.	19.0513	19.1253	0.0740	4.6546	4.6367
4.	20.5767	20.6515	0.0748	4.3128	4.2974
5.	24.6511	24.7242	0.0731	3.6084	3.5979
*Fe-Miron*					
1.	8.1090	8.1845	0.0755	10.8942	10.7940
2.	16.7092	16.8312	0.1220	5.3013	5.2632
3.	19.0158	19.1253	0.1095	4.6632	4.6367
4.	20.5077	20.6515	0.1438	4.3272	4.2974
5.	24.5698	24.7242	0.1544	3.6202	3.5979
*Floridax*					
1.	8.0715	8.1845	0.1130	10.9447	10.7940
2.	16.6862	16.8312	0.1450	5.3086	5.2632
3.	18.9824	19.1253	0.1429	4.6713	4.6367
4.	20.4743	20.6515	0.1772	4.3342	4.2974
5.	24.5410	24.7242	0.1832	3.6244	3.5979
*Body Max*					
1.	8.1790	8.1845	0.0055	10.8011	10.7940
2.	16.6440	16.8312	0.1872	5.3220	5.2632
3.	19.2575	19.1253	0.1322	4.6052	4.6367
4.	20.5430	20.6515	0.1085	4.3198	4.2974
5.	24.8528	24.7242	0.1286	3.5796	3.5979
*Beetroot forte*					
1.	8.0723	8.1845	0.1122	10.9437	10.7940
2.	16.6684	16.8312	0.1628	5.3142	5.2632
3.	18.9598	19.1253	0.1655	4.6768	4.6367
4.	20.4960	20.6515	0.1555	4.3296	4.2974
5.	24.6117	24.7242	0.1125	3.6141	3.5979

* ds—dietary supplement, ** ig—iron (II) gluconate.

**Table 3 molecules-27-00197-t003:** Comparison of experimental data of dietary supplements containing iron (II) fumarate with the experimental data for pure iron (II) fumarate.

No. of Diffraction Line	*2θ* (°) ds *	*2θ* (°) if **	|*∆2θ*|	*d_hkl_* (Å) ds *	*d_hkl_* (Å) if **
*Ferr C Prenacaps*					
1.	11.8235	11.8775	0.0540	7.4787	7.4448
2.	26.3725	26.3244	0.0481	3.3767	3.3827
3.	30.3764	30.3670	0.0094	2.9401	2.9410
4.	33.2966	33.2770	0.0196	2.6886	2.6902
*Ferr C Bicaps*					
1.	11.7865	11.8775	0.0910	7.5021	7.4448
2.	26.3100	26.3244	0.0144	3.3846	3.3827
3.	30.3030	30.3670	0.0640	2.9471	2.9410
4.	33.2574	33.2770	0.0196	2.6917	2.6902
*F-Ferr C*					
1.	11.7282	11.8775	0.1493	7.5393	7.4448
2.	26.2151	26.3244	0.1093	3.3966	3.3827
3.	30.4416	30.3670	0.0746	2.9340	2.9410
4.	33.2121	33.2770	0.0649	2.6953	2.6902
*Iron APTEO*					
1.	11.7697	11.8775	0.1078	7.5128	7.4448
2.	26.2871	26.3244	0.0373	3.3875	3.3827
3.	30.2361	30.3670	0.0067	2.9534	2.9410
4.	33.2637	33.2770	0.1309	2.6912	2.6902
*Restonum LS*					
1.	11.7953	11.8775	0.0802	7.4965	7.4448
2.	26.3718	26.3244	0.0474	3.3768	3.3827
3.	33.3390	33.2770	0.0620	2.6853	2.6902
*Organic iron Aliness*					
1.	11.8105	11.8775	0.0670	7.4869	7.4448
2.	26.3400	26.3244	0.0156	3.3808	3.3827
3.	30.3050	30.3670	0.0620	2.9469	2.9410
4.	33.3355	33.2770	0.0585	2.6856	2.6902
*Organic iron Naturell*					
1.	11.8646	11.8775	0.0129	7.4529	7.4448
2.	26.3226	26.3244	0.0018	3.3830	3.3827
3.	30.3218	30.3670	0.0452	2.9453	2.9410
4.	33.3061	33.2770	0.0291	2.6879	2.6902

* ds—dietary supplement, ** if—iron (II) fumarate.

**Table 4 molecules-27-00197-t004:** Comparison of experimental data of dietary supplements containing iron (II) bisglycinate with the experimental data for pure iron (II) bisglycinate.

No. of Diffraction Line	*2θ* (°) ds *	*2θ* (°) ib **	|*∆2θ*|	*d_hkl_* (Å) ds *	*d_hkl_* (Å) ib **
*Iron Plus*					
1.	16.7089	16.5449	0.1640	5.3014	5.3536
2.	22.6083	22.6979	0.0896	4.3063	3.9143
*Iron Complex NOW*					
1.	16.4626	16.5449	0.0823	5.3802	5.3536
2.	22.7372	22.6979	0.0393	3.9077	3.9143
*Szelazo SR*					
1.	16.5757	16.5449	0.0308	5.3437	5.3536
2.	22.6383	22.6979	0.0596	3.9245	3.9143
*X-Ferr*					
1.	16.7123	16.5449	0.1674	5.3004	5.3536
2.	22.6552	22.6979	0.0427	3.9216	3.9143
*Ferrum DOZ*					
1.	16.6731	16.5449	0.1282	5.3127	5.3536
2.	22.7328	22.6979	0.0349	3.9084	3.9143
*Chella Ferr Med*					
1.	22.6303	22.6979	0.0676	3.9259	3.9143
*Gentle Iron*					
1.	22.5261	22.6979	0.1718	3.9438	3.9143
*Chella Flex*					
1.	16.5961	16.5449	0.0512	5.3372	5.3536
2.	22.5570	22.6979	0.1409	3.9385	3.9143

* ds—dietary supplement, ** ib—iron (II) bisglycinate.

**Table 5 molecules-27-00197-t005:** Parameters determined from TG and DSC analysis for pure iron compounds: iron (II) fumarate, iron (II) gluconate, iron (III) pyrophosphate and iron (III) citrate.

Name of Compound	Weight Loss (%)	Onset (°C)	Endset (°C)	Peak Maximum (°C)	Peak Height (mW)	Peak Area (J)	Enthalpy (J/g)
Iron (II) fumarate	54.9	441.8	473.2	460.5	7.51	2.15	199.3
Iron (II) gluconate monohydrate	54.9	129.5 173.9	150.1 202.6	138.8 189.9	6.65 1.58	1.65 0.45	191.7 52.33
Iron (II) gluconate dihydrate	32.7	139.8 184.3	161.7 205.3	148.3 195.3	5.30 3.13	0.98 0.36	140.7 51.76
Iron (III) pyrophosphate	21.6	266.0	288.2	275.0	0.76	0.09	42.33
Iron (III) citrate	55.5	195.7 304.3	212.9 345.5	201.9 332.2	5.76 1.48	1.17 0.31	87.05 23.47

**Table 6 molecules-27-00197-t006:** Parameters determined from TG and DSC analysis for drugs containing. iron (II) gluconate.

Name of Drug	Weight Loss (%)	Onset (°C)	Endset (°C)	Peak Maximum (°C)	Peak Height (mW)	Peak Area (J)	Enthalpy (J/g)
*Ascofer*	56.6	57.17 124.3 180.1	59.74 153.0 197.4	58.42 140.4 193.6	0.60 7.70 0.99	0.02 2.37 0.15	1.240 175.9 11.02
*Fe-Miron*	63.0	57.21 113.3 167.0	62.13 148.9 176.6	59.11 137.6 171.5	3.56 2.22 1.06	0.16 0.40 0.10	14.66 36.55 8.783
*Ferrous gluconate* *Puritan’s Pride*	53.6	127.6 175.0	144.8 190.0	137.0 181.5	6.07 0.92	1.14 0.06	190.6 10.57

**Table 7 molecules-27-00197-t007:** Parameters determined from TG and DSC analysis for drugs containing iron (II) fumarate.

Name of Drug	Weight Loss (%)	Onset (°C)	Endset (°C)	Peak Maximum (°C)	Peak Height (mW)	Peak Area (J)	Enthalpy (J/g)
*Ferr C Bicaps*	49.9	152.7	173.6	169.0	2.50	0.63	142.5
*Restonum LS*	61.81	90.26 216.2 309.4 388.5	98.67 248.2 312.3 421.2	95.20 238.6 322.1 401.4	3.62 2.52 1.06 0.59	0.29 0.68 0.22 0.13	64.60 151.5 49.99 28.40
*Organic Iron* *Aliness*	62.4	53.50 79.50 142.6 337.2 439.5	68.21 88.23 151.6 414.4 460.4	59.25 83.87 148.3 406.3 450.7	2.69 1.58 1.87 0.86 0.52	0.23 0.16 0.19 0.43 0.07	35.56 24.67 28.72 65.22 10.92

**Table 8 molecules-27-00197-t008:** Parameters determined from TG and DSC analysis for drugs containing iron (III) pyrophosphate.

Name of Drug	Weight Loss (%)	Onset (°C)	Endset (°C)	Peak Maximum (°C)	Peak Height (mW)	Peak Area (J)	Enthalpy (J/g)
*Actiferol*	43.3	81.0 146.2 198.3 237.0	87.0 191.5 204.1 242.6	84.82 175.7 200.4 239.3	6.76 5.26 11.5 2.74	0.38 1.88 0.54 0.27	44.32 218.7 62.74 31.09
*Dicofer Junior*	77.6	87.19 284.3	98.78 290.2	94.31 287.0	7.12 4.77	1.25 1.27	104.1 105.5
*SiderAl Folic*	79.2	87.44 255.4	98.90 309.6	93.56 303.7	4.39 3.42	0.60 2.91	99.25 485.0
*SiderAl Forte*	48.8	165.6 226.6	177.2 263.6	168.6 241.0	1.95 0.466	0.65 0.07	54.22 5.226

**Table 9 molecules-27-00197-t009:** Parameters determined from TG and DSC analysis for drugs containing iron (II) bisglycinate and iron (III) citrate.

Name of Drug	Weight Loss (%)	Onset (°C)	Endset (°C)	Peak Maximum (°C)	Peak Height (mW)	Peak Area (J)	Enthalpy (J/g)
*Chella Ferr Forte*	56.7	99.50 204.6	174.6 267.9	126.0 225.7	0.579 0.540	0.22 0.21	40.06 38.86
*Chella Ferr Med*	59.2	109.2 194.4	143.7 249.7	125.4 211.7	0.983 0.828	0.30 0.32	40.63 43.46
*Chella Flex*	50.3	141.5 332.1 372.1	156.7 339.5 382.6	148.8 330.0 376.3	10.85 0.765 3.933	1.37 0.06 1.00	137.0 6.302 99.95
*X-Ferr*	55.2	143.2	165.5	155.4	0.967	0.08	11.32
*Gentle Iron*	56.5	92.30 198.0	159.0 251.4	126.4 229.0	1.621 1.467	0.73 1.23	52.19 87.56
*Iron NOW*	60.8	99.60 204.7	167.1 260.6	137.5 229.3	1.663 0.664	0.80 0.21	66.87 17.17
*Iron citrate Swanson*	45.8	268.7 312.8	312.3 326.5	285.0 321.8	2.092 1.328	0.85 0.17	206.6 40.85

**Table 10 molecules-27-00197-t010:** Analyzed dietary supplements containing iron compounds.

No.	Product Name (*Manufacturer*)	Form of Iron	Iron Compound Content in 1 Tablet/Sachet [mg]	Iron Compound Content in 1 Tablet/Sachet (%Weight)
1.	*Ascofer* (ESPEFA)	Iron (II) gluconate	200.0	78.4
2.	*Fe-miron* (Domowa apteczka)	Iron (II) gluconate	64.7	21.6
3.	*Beetroot forte* (Herbapol)	Iron (II) gluconate	12.0	1.78
4.	*BodyMax* (ORKLA CARE A/S)	Iron (II) gluconate	144.4	3.21
5.	*Floradix* (Salus)	Iron (II) gluconate	112.3	24.4
6.	*Ferrous gluconate* (Puritan’s Pride)	Iron (II) gluconate	172.4	47.5
7.	*Iron* (APTEO)	Iron (II) fumarate	76.8	18.7
8.	*F-Ferr C* (ForMeds)	Iron (II) fumarate	42.0	5.83
9.	*Restonum LS* (Aflofarm)	Iron (II) fumarate	61.4	6.61
10.	*Organic Iron* (Naturell)	Iron (II) fumarate	86.0	18.3
11.	*Organic Iron* (Aliness)	Iron (II) fumarate	76.8	12.5
12.	*Ferr C Prenacaps* (ForMeds)	Iron (II) fumarate	86.0	14.3
13.	*Ferr C Bicaps* (ForMeds)	Iron (II) fumarate	86.0	16.3
14.	*Actiferol Fe Forte* (Polski Lek)	Iron (III) pyrophosphate	200.0	37.4
15.	*Dicofer Junior* (VITIS PHARMA)	Iron (III) pyrophosphate	100.0	6.25
16.	*SiderAl Folic* (Pharma Natura)	Iron (III) pyrophosphate	70.0	4.38
17.	*SiderAl Forte* (Pharma Natura)	Iron (III) pyrophosphate	200.0	34.2
18.	*Smart Hit IV Ferrum* (Valentis)	Iron (III) pyrophosphate	93.4	16.7
19.	*Biofer* (ORKLA CARE A/S)	Iron (II) bisglycinate	72.3	8.82
20.	*Ferrum* (DOZ Product)	Iron (II) bisglycinate	150.0	2910
21.	*Solar Gentle Iron* (Solgar Vitamins and Herbs)	Iron (II) bisglycinate	103.1	22.2
22.	*Chella Ferr Forte* (Olimp)	Iron (II) bisglycinate	150.0	36.1
23.	*Chella Ferr Med* (Olimp)	Iron (II) bisglycinate	160.7	38.7
24.	*Chella Ferr Bio* (Olimp)	Iron (II) bisglycinate	75.0	18.75
25.	*Chella Flex Iron* (Oleofarm)	Iron (II) bisglycinate	150.0	29.1
26.	*Szelazo + SR* (Lek-AM)	Iron (II) bisglycinate	150.0	17.9
27.	*Iron Plus* (CaliVita)	Iron (II) bisglycinate	14.0	8.75
28.	*Balanced Iron Complex* (Viridian Nutrition)	Iron (II) bisglycinate	80.4	15.4
29.	*Iron Complex* (Nutricion for Optimal Wellness)	Iron (II) bisglycinate	96.4	20.0
30.	*X-Ferr* (Galena)	Iron (II) bisglycinate	150.0	16.8
31.	*Iron citrate* (Swanson)	Iron (III) citrate	110.0	28.0

**Table 11 molecules-27-00197-t011:** List of used ICDD PDF4 cards.

No.	Chemical Compound	Chemical Formula	No. PDF Card
1.	Iron (II) fumarate	C_4_H_2_FeO_4_	00-062-1294
2.	Ascorbic acid	C_6_H_8_O_6_	00-004-0308
3.	Cellulose	(C_6_H_10_O_5_)_x_	00-003-0226
4.	Sorbitol	C_6_H_14_O_6_	00-027-1643
5.	Titanium (IV) oxide	TiO_2_	00-015-0875

**Table 12 molecules-27-00197-t012:** Summary of results of both methodologies.

Dietary Supplements	Summary	
	XRPD	DSC/TG
Containing crystalline iron compounds: Iron (II) gluconate Iron (II) fumarate	Positions of diffraction lines (angle values *2θ*) of iron compounds compatible with data presented in the ICDD database. Values of *Δ2θ* < 0.2°—confirm the presence of the proper iron compound in the investigated samples.	Resolution of hydrates with various numbers of H_2_O. Confirmation of the presence of iron (II) gluconate monohydrate. Confirmation of the presence of iron (II) fumarate. Determination decomposition temperature.
Containing semicrystalline iron compounds: Iron (II) bisglycinate	Possible to the identification of the presence of iron (II) bisglycinate. XRPD diffraction of pure iron (II) bisglycinate revealed two diffraction lines. Differences in diffraction line positions between pure iron (II) bisglycinate and dietary supplements (*Δ2θ*) < 0.2°—confirmation of the presence of this compound in the investigated samples.	A different way of sample decomposition depends on the crystal components present. Exothermic peaks are visible from the amorphous substances present on the curves. Crystallization process likely. Determination of decomposition temperature.
Containing amorphous iron compounds: Iron (III) citrate Iron (III) pyrophosphate	Lack of visible diffraction lines. Presence of amorphous “halo”.	Determination of decomposition characteristics. Determination of destroyed temperatures.

## Data Availability

The data presented in this study are available on request from the corresponding author.

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
