# Peer review of "The Usefulness of X-ray Diffraction and Thermal Analysis to Study Dietary Supplements Containing Iron"

_molecules, 2021, doi:10.3390/molecules27010197_

Round 1
Reviewer 1 Report
The aim of the manuscript is a fast and sufficient analysis of substances in food supplements, that contain iron in different forms. The food supplements are in the border of pharmacy and food industry. The analytical technic used for qualitative analysis was XRPD and thermic methods (DSC TGA). The efficiency of finding the specific compound containing iron (II or III) in the formulation is questionable. It seems that it works well for crystalline and semicrystalline compounds, but the amorphous compound seems to be hard to recognize in the formulation. The conclusion should be written less strongly, and the possibility of trouble with amorphous compounds should be mentioned. Point 1) of major revision will help with the clarification of the usefulness of the analytical techniques and their combination.
Major points:
1) Adding a table with a summary of observations of individual food supplements (using both analytical techniques) and its discussion will be helpful with clarification of the manuscript.
2) Conclusion and abstract should be rewritten concerning troubles with amorphous compounds, which are highly problematic in most of the analytical techniques used in the pharma industry.
Minor points:
1) Name of the article: The X-ray should be extended to X-ray diffraction to be clear which type of technic was investigated
2)" X-ray phase analysis of iron compounds": Are there also single-crystal structures? Is it known also powder pattern of iron (II) gluconate dihydrate?
3) Figure 1 iron (II) gluconate is monohydrate or dihydrate?
4) Figure 7. overlapping of numbers
5) Table 1: The density of iron fumarate is strange. Need to be checked.
6) Powder pattern could be clearer if the shift of the base will be used
Author Response
We would like to thank the Reviewer for carefully examining our paper.
The manuscript was carefully revised and adapted according to the reviewer’s notes and comments. All changes in the manuscript are highlighted in green.
Review Report Form I
English language and style: (x) English language and style are fine/minor spell check required
Comments and Suggestions for Authors
The aim of the manuscript is a fast and sufficient analysis of substances in food supplements, that contain iron in different forms. The food supplements are in the border of pharmacy and food industry. The analytical technic used for qualitative analysis was XRPD and thermic methods (DSC TGA). The efficiency of finding the specific compound containing iron (II or III) in the formulation is questionable. It seems that it works well for crystalline and semicrystalline compounds, but the amorphous compound seems to be hard to recognize in the formulation. The conclusion should be written less strongly, and the possibility of trouble with amorphous compounds should be mentioned.
Point 1) of major revision will help with the clarification of the usefulness of the analytical techniques and their combination.
Major points:
1) Adding a table with a summary of observations of individual food supplements (using both analytical techniques) and its discussion will be helpful with clarification of the manuscript.
A table summarizing the observations has been inserted in the Conclusion.
2) Conclusion and abstract should be rewritten concerning troubles with amorphous compounds, which are highly problematic in most of the analytical techniques used in the pharma industry.
The information concerning troubles with amorphous compounds has been added to the text.
Abstract and Conclusion have been rewritten.
Minor points:
1) Name of the article: The X-ray should be extended to X-ray diffraction to be clear which type of technic was investigated
The title has been corrected.
2)" X-ray phase analysis of iron compounds": Are there also single-crystal structures? Is it known also powder pattern of iron (II) gluconate dihydrate?
We used the X-ray powder diffraction method. All investigated iron compounds were in polycrystalline form.
In the manuscript of J.W. Reid entitled “Powder diffraction data of ferrous gluconate”, we can find the general formula of ferrous gluconate as C12H24FeO14·xH2O. The author did not indicate clearly what is the value x. The explanation is written below.
3) Figure 1 iron (II) gluconate is monohydrate or dihydrate?
In Figure 1, the iron (II) gluconate monohydrate is presented (it has been corrected). As we mentioned in the text, the diffraction pattern of both forms (monohydrate and dehydrate) are almost identical, the XRD method does not distinguish the form of hydrates.
4) Figure 7. overlapping of numbers
It has been corrected.
5) Table 1: The density of iron fumarate is strange. Need to be checked.
Thank you for paying attention to calculation mistake. The correct iron (II) fumarate density value has been inserted into the table.
6) Powder pattern could be clearer if the shift of the base will be used
Figures showing X-ray images have been corrected as suggested by the reviewer.

Reviewer 2 Report
Thank you for inviting to review the manuscript “The useful of X-ray and thermal analysis to study dietary supplements containing iron”. The study has some interesting findings on use of X-ray and thermal analysis for phase composition determination in micronutrient supplements, and detection of polymorphism in in Fe. The results presented are encouraging and promising however there are many areas that needs attention of the authors and have a lot of room for improvement.
Revisit the abstract to modify the question and purpose of the study, summarize the main findings only in results. Conclusion statements are too broader and not substantiated with the data presented.
I am not convinced with the idea of presenting literature review that is quite deviating from the scope/ background and objective of the study. Background of the study given provide extensive details on health significance of various forms of iron and the health risks while in actual the data presented looks promising as a method development to study Fe containing dietary supplements.
The results section is more elaborative and carries unnecessary details of the methods making it difficult to locate the text / data that relates to present study. At an instant check the sub-section 2.1. I would suggest avoiding methods explanation in results e.g., line 269-274.
Some other minor comments are given below:-
Line 78: “high consumption of red meat, has been connected with the risk of diabetes or cancer development”. The citation given looks irrelevant. Instead, association of high heme iron intake through red meat consumption has been reported by Ward, M. H., Cross, A. J., Abnet, C. C., Sinha, R., Markin, R. S., & Weisenburger, D. D. (2012). Heme iron from meat and risk of adenocarcinoma of the esophagus and stomach. European Journal of Cancer Prevention, 21(2), 134.
Line 92: I suggest to revise as “comply with the conditions mentioned above”
Line 444-446: Remove this text
Line 358: I would suggest a revision like “at a temperature close to one for the pure iron fumarate i.e., 450℃.
Line 374-376: You may add some explanation to thermal decomposition of the tested samples at different temperature and reasoning for the one decomposed at 290℃.
Conclusions are not supporting the outcomes of project, too lengthy and non-directional. I could not find or maybe I missed, the minimum detectability limits of the methods if presented? Please check the statement at line 498.
Line 498-503: I suggest revising this section at all. Recommendations are too straightforward and may contradict with the findings of the study
Author Response
We would like to thank the Reviewer for carefully examining our paper.
The manuscript was carefully revised and adapted according to the reviewer’s notes and comments. All changes in the manuscript are highlighted in green.
Review Report Form II
English language and style: (x) I don't feel qualified to judge about the English language and style
Comments and Suggestions for Authors
Thank you for inviting to review the manuscript “The useful of X-ray and thermal analysis to study dietary supplements containing iron”. The study has some interesting findings on use of X-ray and thermal analysis for phase composition determination in micronutrient supplements, and detection of polymorphism in in Fe. The results presented are encouraging and promising however there are many areas that needs attention of the authors and have a lot of room for improvement.
Revisit the abstract to modify the question and purpose of the study, summarize the main findings only in results. Conclusion statements are too broader and not substantiated with the data presented.
Abstract and Conclusion have been rewritten.
I am not convinced with the idea of presenting literature review that is quite deviating from the scope/ background and objective of the study. Background of the study given provide extensive details on health significance of various forms of iron and the health risks while in actual the data presented looks promising as a method development to study Fe containing dietary supplements.
Dear reviewer, thank you for this valuable opinion. Our purpose was to describe and underline the complex situation observed in the market and the use of iron supplements. Lack of proper control of iron supplements that are freely available to the patients makes them especially menacing. Potential harm from improper use or abuse of iron supplements is higher than in other supplements consisting of trace metals or vitamins. On the other hand, there is little or no data about the impact of formulation on stability, bioavailability or potential toxicity of iron compounds used as supplements. At the same time, supplementation of iron is recommended by most governmental and health authorities, including WHO. We believe this situation should be discussed and needs continuous efforts with analytical aspects.
The introduction has been rewritten, shortened and the references modified.
The results section is more elaborative and carries unnecessary details of the methods making it difficult to locate the text / data that relates to present study. At an instant check the sub-section 2.1. I would suggest avoiding methods explanation in results e.g., line 269-274.
Some other minor comments are given below:
Line 78: “high consumption of red meat, has been connected with the risk of diabetes or cancer development”. The citation given looks irrelevant. Instead, association of high heme iron intake through red meat consumption has been reported by Ward, M. H., Cross, A. J., Abnet, C. C., Sinha, R., Markin, R. S., & Weisenburger, D. D. (2012). Heme iron from meat and risk of adenocarcinoma of the esophagus and stomach. European Journal of Cancer Prevention, 21(2), 134.
References have been changed.
Line 92: I suggest to revise as “comply with the conditions mentioned above”
Thank you, we revised this sentence.
Line 444-446: Remove this text
The text on these lines has been removed.
Line 358: I would suggest a revision like “at a temperature close to one for the pure iron fumarate i.e., 450℃.
It has been corrected.
Line 374-376: You may add some explanation to thermal decomposition of the tested samples at different temperature and reasoning for the one decomposed at 290℃.
The explanation has been added to the text.
Conclusions are not supporting the outcomes of project, too lengthy and non-directional. I could not find or maybe I missed, the minimum detectability limits of the methods if presented? Please check the statement at line 498.
We mentioned about X-ray detection limit in lines 161-164. The X-ray detection limit (LOD) is 0.1 – 1%wg. As seen in Table 11, the amount of iron compounds is larger than 1%wg. For this reason, X-ray diffraction is a proper method to identify the phase composition of these dietary supplements.
However, it is worth mentioning that limit of quantification (LOQ) depends on many factors: particle size, preferred orientation, line overlap, crystal symmetry and amorphous substances. This problem is described in [26].
Line 498-503: I suggest revising this section at all. Recommendations are too straightforward and may contradict with the findings of the study
The Conclusions have been corrected.
Round 2
Reviewer 1 Report
I would like thanks to authors for their effort in improving the manuscript, it helped a lot. But, there are still things which could be changed.
I would like to apologize because I wrongly described the table in my first point in the last round of the review. It should be more specific to this particular study. From my point of view, the best table is with four columns (1)Name of supplement (2)iron compound in the supplement (3) XRPD (proved / unclear) (4) thermic methods (proved / unclear). It makes clear usability the method is for this particular study.
Major point:
Fig 3.: Body Max pattern does not fit to iron (ii) gluconate or the diffraction of the iron (ii) gluconate is too weak.
Minor point:
Line 400/401 duplicity (on examination on the examination)
Tab 1: there should be written deviations of cell parameters and 90.00°angles should be rounded to 90°.
Fig 3. Ascofer should be corrected (zero shift correction)
Fig 3b: Why there is not the red pattern (iron gluconate)in the figure?
Fig. 4.: Restornum LS should be corrected (zero shift correction)
Author Response
We would like to thank the Reviewer for carefully examining our paper.
The manuscript was carefully revised and adapted according to the reviewer’s notes and comments. All changes in the manuscript are highlighted in yellow.
Review Report Form 1
English language and style: (x) English language and style are fine/minor spell check required
Comments and Suggestions for Authors
I would like thanks to authors for their effort in improving the manuscript, it helped a lot. But, there are still things which could be changed.
I would like to apologize because I wrongly described the table in my first point in the last round of the review. It should be more specific to this particular study. From my point of view, the best table is with four columns (1)Name of supplement (2)iron compound in the supplement (3) XRPD (proved / unclear) (4) thermic methods (proved / unclear). It makes clear usability the method is for this particular study.
Response: The authors believe that such a large table does not fit into the Conclusion. Therefore, they allowed themselves to keep Table 12. However, the authors modified the overall conclusion so as to be as close as possible to the reviewer's proposal and summarized the data he requires in the text of the conclusion.
Major point:
Fig 3.: Body Max pattern does not fit to iron (ii) gluconate or the diffraction of the iron (ii) gluconate is too weak.
Response: Modified according to the reviewer's proposal.
Minor point:
Line 400/401 duplicity (on examination on the examination)
Response: Duplicity has been removed.
Tab 1: there should be written deviations of cell parameters and 90.00°angles should be rounded to 90°.
Response: Modified according to the reviewer's proposal.
Fig 3. Ascofer should be corrected (zero shift correction)
Response: Modified according to the reviewer's proposal.
Fig 3b: Why there is not the red pattern (iron gluconate) in the figure?
Response: This supplement contains amorphous substances that have significantly elevated the background of the measurement. Moreover, the low intensity of diffraction lines is confirmed by the small amount of iron gluconate in the tested supplement. A drawing showing additionally crystalline iron gluconate would be unreadable
Fig. 4.: Restornum LS should be corrected (zero shift correction)
Response: Modified according to the reviewer's proposal.
Reviewer 2 Report
Thank you for accepting suggestions and submitting a revised version of the manuscript.
After going through the revised version of the manuscript, I feel confident to comment that authors have satisfactorily addressed all queries and suggestions. I would like to recommend the revised version to be considered for publication in Molecules.
Author Response
Response: The authors thank the reviewer for appreciating the manuscript.